# A Core-Shell Approach for Systematically Coarsening Nanoparticle–Membrane Interactions: Application to Silver Nanoparticles

**DOI:** 10.3390/nano12213859

**Published:** 2022-11-01

**Authors:** Ankush Singhal, G. J. Agur Sevink

**Affiliations:** Leiden Institute of Chemistry, Leiden University, P.O. Box 9502, 2300 RA Leiden, The Netherlands

**Keywords:** nanotoxicity, simulation, molecular dynamics (MD), coarse-grained, systematic, nanoparticle, lipid bilayer, core-shell

## Abstract

The continuous release of engineered nanomaterial (ENM) into the environment may bring about health concerns following human exposure. One important source of ENMs are silver nanoparticles (NPs) that are extensively used as anti-bacterial additives. The introduction of ENMs into the human body can occur via ingestion, skin uptake or the respiratory system. Therefore, evaluating how NPs translocate over bio-membranes is essential in assessing their primary toxicity. Unfortunately, data regarding membrane–NP interaction is still scarce, as is theoretical and in silico insight into what governs adhesion and translocation for the most relevant NPs and membranes. Coarse-grained (CG) molecular descriptions have the potential to alleviate this situation, but are hampered by the absence of a direct link to NP materials and membrane adhesion mechanisms. Here, we interrogate the relationship between the most common NP representation at the CG level and the adhesion characteristics of a model lung membrane. We find that this representation for silver NPs is non-transferable, meaning that a proper CG representation for one size is not suited for other sizes. We also identify two basic types of primary adhesion—(partial) NPs wrapping by the membrane and NP insertion into the membrane—that closely relate to the overall NP hydrophobicity and significantly differ in terms of lipid coatings. The proven non-transferability of the standard CG representation with size forms an inspiration for introducing a core-shell model even for bare NPs that are uniform in composition. Using existing all-atom molecular dynamics (MD) data as a reference, we show that this extension does allow us to reproduce size-dependent NP adhesion properties and lipid responses to NP binding at the CG level. The subsequent CGMD evaluation for 10 nm Ag NPs provides new insight into membrane binding for relevant NP sizes and into the role of water in trapping NPs into defected mixed monolayer–bilayer states. This development will be instrumental for simulating NP–membrane adhesion towards more experimentally relevant length and time scales for particular NP materials.

## 1. Introduction

Owing to the tunability of their physical appearance, properties and responses, nanoparticles (NPs) have acquired a critical position in various areas of application, including cancer therapeutics, imaging [1], drug delivery [2], cosmetics [3], catalysis and as low-cost functional additives in food, health care and textile fields [3,4]. While they are so widely used, detailed information on how NPs interact with the structured bio-matrix that surrounds them, other than phenomenological, is surprisingly sparse. In particular, while genuine limitations to the spatial and temporal resolution of current characterization methods seriously limit our understanding of interactions and mechanisms at this bio-interface from direct or indirect observations, in silico study is equally hindered by the reductionism that is inherent to most theoretical descriptions as well as the computational demands of atomically resolved molecular modelling. Yet, understanding the link between the design space for engineered NP and the complex bio-matrix that such NPs are (eventually) released in on one side and toxicity of some form, including nanotoxicity induced by the release of ions, the production of oxidative species and NP translocation over bio-membranes and subsequent interference with metabolic pathways on the other side, is of paramount importance. Focusing on NP uptake, one amenable way of increasing this understanding is by employing computationally more efficient molecular modelling approaches that are capable of sampling many system variables with sub-molecular resolution, including variables like the surface ligand composition and distribution, nanoparticle morphology and lipid membrane composition.

Previously, theoretical descriptions that consider the membrane as an elastic sheet related the uptake of NPs to a subtle balance of the energy needed for the membrane to deform, reflected in the membrane bending rigidity, lateral tension, and the energetic gain of NP adsorption, provided by the adhesion energy density of (coated) NPs in the contact region and, as was later concluded, also in non-contact regions [5,6,7]. It is therefore clear that, in order for any computational procedure to go beyond these effective models, at least these two attributes should be adequately reproduced. While much effort has gone into the systematic coarsening of lipids towards proper reproduction of the membrane attributes, [8,9,10,11] the NP side has received considerably less attention. Here, we focus on calibrating a frequently used coarse-grained representation, i.e., the Martini model, towards accurately representing both aspects simultaneously, in order to enable a seamless description of multicomponent NP–membrane interactions from the all-atom to the molecular fragment scale. It should be noted that molecular detail is particularly desirable over the continuum representation in an elastic model for multi-component membranes, since Lunnoo et al. demonstrated for gold NPs that the lipid composition of a bilayer plays a pivotal role in NP uptake [12]. Enabling simulation of larger NPs than those that are within reach of atomistic simulation methods is of importance, as size has been shown to play a significant role in the binding mechanisms. These can take the form of NP adsorption onto the lipid membrane (bilayer-wrapped) [13], NP internalization within the lipid bilayer (internalized, i.e., monolayer) [14] and full NP engulfment [15]. Internalization of NPs has previously been demonstrated in the absence of receptor-mediated uptake, [16,17] indicative of the role of non-specific interactions. In this study, we will show that systematic coarsening is needed for reproducing proper binding kinetics and that it is advantageous for extracting size-dependent NP properties. Previously, similar agreement between all-atom (AA) and coarse-grained (CG) MD results was constituted for lipids, polymers [18,19], carbon-based nanoparticles [20,21], and ligated quantum dots [22].

Mapping systems from the atomistic to the coarse grained (CG) domain can be strenuous, since one attempts to project relevant thermodynamic properties to a description that has a (strongly) reduced number of degrees of freedom. Even if successful, one should be aware of the general issues associated with coarse graining [23], which include the understanding that the resulting map is usually only valid close to the considered thermodynamic state [24]. The chosen CG Martini (version 2) approach considers a hybrid mapping scheme in which all bonded interactions between CG beads are obtained via systematic inversion of atomistic data. The strength ϵ of the non-bonded Lennard–Jones (LJ) interactions, on the other hand, is mapped from (experimental) thermodynamic data that is provided as the partitioning free energies for a limited number of CG particle types between water and organic solvent phases [8]. As a result, Martini can reasonably well predict binding and insertion free energies for small molecules like peptides in one-component membranes [25]. Yet, for large solid objects like nanoparticles, the Martini parametrization scheme for beads making up the NP, i.e., obtained from bead partitioning in fluid domains, is not necessarily accurate. It is particularly known that the interactions between fluid (lipid) domains and solid objects are dominated by special conditions at the fluid–solid interface, which can be separated into entropic (reduced conformational space) and enthalpic (effective surface potential) contributions. While there is a challenge for both contributions to be mapped separately and accurately from the atomistic level, a separation of the NP into a *core* and *shell* domain, made up of different existing Martini bead types, provides a useful handle for tuning the surface potential via the choice of available bead types for these two separate domains. Avoiding the need for introducing new CG bead types has a clear advantage that no general reparametrization of Martini is required, while the 2-type core-shell representation avoids any issues of overfitting that would hamper even more heterogeneous (3, 4, 5, etc., bead types) NP representations. We will later argue that a separation in core and shell is also physically appropriate. The ability to tune the effective surface potential via a core-shell description, where the shell domain in their case represents a NP ligand coating, was previously illustrated by Gupta et al., where the nature of the NP, represented by the core domain, was fixed. They observed that the surface chemistry, i.e., the nature of the ligands, has a profound effect on the free energy profile of NP insertion and thus on the translocation pathway of the NP [26]. Moreover, another study previously exploited this separation, with the NP surface parametrized at an atomistic scale and the core at a continuum level [27].

Here, we fundamentally consider this CG *core-shell* representation for the first time for bare NPs. Considering that previous studies used a core-shell model for ligated NPs, we note that, owing to the limited spatial extent of the effective surface potential, the interactions for coated NPs may be dominated by the ligands, as suggested by the work of the Rossi group for small gold NPs coated with hydrophobic and hydrophilic (charged) ligands [22]. In our setup, relevant Martini bead types for the core-shell model are determined by matching the atomistic and CG potential of mean forces (PMFs) for insertion, since these free energies directly relate to binding energies and membrane perturbation characteristics. In particular, our (manual) optimization procedure considers multiple objectives, since we require the selected core-shell parameters to be transferable between NPs of different sizes. For comparison, we will relate our results also to the usual assignment of bead types in terms of chemical fragments, in which case all NP beads are selected to be of the same Martini type. These two options are illustrated in Figure 1A.

The protocol developed in the study is generally applicable, but we illustrate it for a material that is extensively applied in NP form for our illustration of its capabilities. Silver (Ag) NPs are known for their biocidal activity due to the release of Ag${}^{+}$ and can, for certain NP surface modifications, also induce oxidative damage [28,29,30]. Their effectiveness as bactericides at nanomolar concentration has led to a growing use of Ag NP as additives in commercial products [31,32], substantially elevating the risk of human exposure by their release into the ecosystem [33,34]. The starting points for our current study are previous constrained and unconstrained all-atom molecular simulations (AAMD) of 1–5 nm NPs [35,36]. Besides corroborating that bare Ag NPs are more hydrophobic than their SiO${}_{2}$ and TiO${}_{2}$ counterparts, Ag NPs of all considered sizes were found to modulate a model lung membrane. In particular, the membrane wraps around the NP, with a binding free energy that increases with increasing diameter [19]. The desire to understand the particulars of this wrapping as a function of NP size and hydrophobicity—see Figure 1B for the two observed mechanisms for smaller NPs—and the recent suggestion of a discontinuous wrapping transition for 10–15 nm NPs from a coarser CG model than considered here [37], warrant an investigation into how our Martini-based CGMD can be related to the AAMD results for the previously considered reduced NP sizes. We will particularly go beyond the limits of the previous atomistic study by performing microsecond unconstrained CGMD simulation of a lipid bilayer with a 10 nm NP, investigating adhesion characteristics towards realistic NP sizes and the extraction of size-dependent nano-descriptors that are needed for quantitative structure–activity relationship (QSAR) analysis. The next step, exploiting the calibrated CG model for studying the binding energy for NPs ≫ 10 nm as well as for other specific NP materials is left for future consideration. It needs to be noted that to date, no CG force field has been developed for NPs and a complex lipid bilayer that retains the atomistic interaction properties.

## 2. Materials and Methods

Simulations were performed with the GROMACS 2020 package [38] using the Martini 2.0 representation [8]. The initial atomistic structures of Ag NP for sizes 3, 5, and 10 nm were retrieved from CHARMM-GUI builder [39]. We assume that NPs do not change with time, disregarding that silver is quite soluble in reality. Following the usual approach for coarsening NPs—see, for instance, Salassi et al. [22]—each Ag atom was mapped to a single (uncharged) CG bead. The core-shell model considers the first row of surface beads, in direct contact with solvent or lipids, to represent the NP shell domain, while the rest is considered to be part of the core domain; see Figure 1A. To maintain the (initial) spherical NP geometry, Ag CG beads were constrained with a harmonic potential having a bond constant value, K${}_{Bond}$ = 1500 kJ mol${}^{-1}$ nm${}^{-2}$ and an equilibrium distance, r${}_{eq}$ = 0.27 nm. A complex quaternary lipid membrane consisting of Dipalmitoylphosphatidylcholine (DPPC), 1,2-dioleoyl-sn-glycerol-3-phosphocholine (DOPC), 1-palmitoyl-2-oleoyl-sn-glycerol-3-phosphocholine (POPC) and cholesterol (CHOL) in ratio 5:2:2:1 as used in previous studies [35,40,41] was considered to represent the model lung membrane. The initial membrane configuration was generated using the INSANE building tool [42]. Two types of simulations were carried out: (1) an Umbrella Sampling (US) technique was employed for calculating the PMFs relating to the insertion of single Ag NPs of 3 and 5 nm in size into the lipid membrane, and (2) unbiased simulation of Ag NP–membrane-solvent systems for sizes of 3, 5 and 10 nm. For visualization purposes, Visual Molecular Dynamics (VMD) [43] was used.

### 2.1. Setup for Unconstrained Coarse Grained Molecular Dynamics

A single NP was manually inserted 2.0 nm above the equilibrated lipid membrane and subsequently solvated with CG water beads (of which 5–10% were antifreeze water beads). The membrane area was taken ∼4.5 times larger that the NP diameter to avoid significant artifacts due to the periodic boundary conditions that we employed throughout this study. A table listing membrane area and compositions for each simulated system can be found in the Appendix A. A 1.4 nm cutoff was chosen for both the electrostatic and Lennard–Jones interactions. A particle mesh Ewald method [44] was used to evaluate long-range electrostatic interaction, with a relative dielectric constant ($\epsilon_{r}$) of 15 for explicit screening. The potential energy of the initial conformations was minimized using the steepest descent algorithm [45], followed by two small equilibration intervals of 10 nanoseconds (ns), with a 1 femtosecond (fs) time step and another 10 ns using a 10 fs time step. Finally, the production run of 10 microseconds (μs) was performed for all considered NP sizes with a time step of 20 fs. It should be noted that the time steps in CGMD do not necessarily relate to real time because of the effect that coarsening has on the kinetics and should be calibrated via known diffusion rates. All the production runs were performed using a v-rescale thermostat [46] at 310 K, while the pressure was controlled with a Parrinello–Rahman barostat [47] in a semi-isotropic setup at 1 bar and a compressibility of β = 3 × 10${}^{-4}$ bar${}^{-1}$. The last 2 μs of production runs were used throughout for the purpose of analysis. The partial density, radial distribution function (g(r)) and mean square displacement (MSD) as a function of time were calculated using the built-in GROMACS routines.

### 2.2. Setup for Potential of Mean Force calculation

We employed Umbrella Sampling (US) [48] to compute the potential of mean force (PMF) for inserting a single nanoparticle into the lipid bilayer, taking the center of mass (COM) distance between both structures as our reaction variable. For the pulling, we used a force constant K = 500 kJ mol${}^{-1}$ nm${}^{-2}$ and an equidistant spacing along the reaction coordinate of 0.125 nm, resulting in 35–45 configurational windows. As we intend to match the CG PMFs to their atomistic counterparts, our PMF calculation for the CG case was limited to 3 and 5 nm Ag NPs. In particular, the computational demands of US become increasingly prohibitive for systems containing much larger NPs, even at the CG level, and can better be replaced by techniques that calculate the binding free energy based only on two states. Slow relaxation can be a concern in US and give rise to improper sampling along a reaction coordinate. In particular, the kinetics of membrane remodelling upon significant perturbation, being an intrinsically slow phenomenon on the time scales of CGMD, can have a residual effect on PMFs. For this reason, we investigated convergence by considering different sampling periods along the 400 ns production run in the calculation of all PMFs, for all configurational windows and for both NP sizes considered. The PMFs calculated using 100, 200, 300 and 400 ns sampling periods generally show a very slight variation in barrier height after 200 ns; see Appendix A. From this, we concluded that 200 ns sampling per window suffices for a properly equilibrated PMF. For all considered US windows, the pressure in the 200 ns npT production run was controlled at 1 bar using a Parinello–Rahman barostat [47] in a semi-isotropic pressure coupling setup. Again, periodic boundary conditions were employed and simulations performed at 310 K using a v-rescale thermostat [46]. The standard weighted histogram analysis method (WHAM) [49,50] was consequently applied to extract the free energy profile. The error associated with the calculation of the PMFs was analyzed using a bootstrap routine that is available within GROMACS. The obtained CG PMFs were compared to the atomistic PMFs taken from our previous work [35].

## 3. Results and Discussion

### 3.1. Establishing a Size-Invariant Martini CG Representation for NPs: The Ag Case

In a recently published AAMD study of 3 and 5 nm NPs, we showed that individual bare Ag NPs bind and perturb the considered model lung membrane, irrespective of their size [35]. This observation is in stark contrast to the findings for bare SiO${}_{2}$ and TiO${}_{2}$ NPs in the same study, which show only weak or no membrane affinity for 3 and 5 nm NPs. In particular, Ag NPs gain a substantial amount of free energy when translocating from the aqueous phase to the liquid crystalline bilayer. Yet, these AAMD simulations also clearly illustrated that binding energies and binding mechanisms can vary significantly with NP size. Owing to the prohibitive computational demands of atomistic modelling for systems that more closely resemble experimental setups, particularly in terms of NP size, which is usually at least 20–50 nm, coarsening is an established need.

On the other hand, retaining the essence of a particular system at a coarser level is also a necessity if one wants to distinguish between actual NPs and membranes in a systematic fashion. The Martini CG representation [8] is one of the most commonly uignaed in CGMDs, owing to its performance, force-field like nature and subsequent ease of use and the extensive development and user community [51]. A critical feature of Martini, which it actually shares with many other coarsening schemes, is a hybrid mapping approach, in which parameters of non-bonded interactions between beads are determined from a top-down thermodynamic matching of water/oil partitioning free energies, while parameters for the bonded interactions are obtained from an iterative inversion of atomistic structural data. In this way, the map considers both thermodynamic properties and atomistic simulation data simultaneously. A special feature is that Martini 2 considers only a limited number (four) of interaction site types, polar (P), non-polar (N), charged (Q) and apolar (C), that are each quantified by their relative strength (from 1 to 5). For instance, beads of the C1 type are more hydrophobic than those of the C5 type.

Historically, each of these CG Martini beads are chosen to represent a chemical (molecular) fragment containing four heavy atoms, but in the past also a number of alternative fragmentation schemes and new bead types have been introduced to address specific needs. The viewpoint that CG Martini beads map a particular chemistry one-to-one, i.e., that each bead represents a chemical fragment that belongs to a distinct class of equivalence, is both simple and attractive. Yet, it is not necessarily a valid viewpoint, as we will show in the remainder of the article.

Here, we will first develop a general and accurate material-specific protocol for mapping a solid Ag nanoparticle from the all-atom (AA) domain to an effective CG Martini representation. To do so, we start by analysing the properties and limitations of the most intuitive standard map, i.e., the one in which all NP atoms are represented by the same type of CG Martini beads. For this *uniform* CG representation, we systematically consider the energetics of binding (via the CG PMFs) and the binding mechanism (via unconstrained CGMD simulation) for all relevant Martini bead types. We will particularly focus on 3 and 5 nm NPs for which we have AA PMFs, i.e., PMFs obtained by AAMD, as a reference. Seeing that AA and CG PMFs for these NP sizes cannot be matched simultaneously, we will analyze CG PMFs for these sizes in the new *core-shell* CG NP representation, in which the bead types of both the core and shell domain can be selected independently. We discuss why this representation, which maps the same atoms to different CG beads, is appropriate and useful. We particularly show that the introduction of different bead types for the core and shell domain does enable matching of AA PMFs and binding mechanisms in all key aspects for both NP sizes. For ease of discussion, we use the same notation for the core-shell and uniform representation, i.e., C${}_{Ci}$-S${}_{Cj}$, with *i* representing the strength of a core *C* of C-type beads and *j* that of the shell *S*, since the latter can be seen as a special case of the first. The resulting CG Martini representation of an Ag NP will subsequently be applied for simulating a 10 nm NP by CGMD.

#### 3.1.1. Analyzing the Standard Uniform NP Representation

In Martini CG, NPs are commonly represented by an aggregate of beads of the *same* CG Martini type, i.e., the uniform representation. For instance, two groups recently performed a detailed study of the interaction between tiny coated gold NPs and pure POPC or mixed POPC/POPG membranes, employing a uniform representation of the gold NP as a collection of C5 beads [22,52]. While they identified a proper qualitative match between CG simulation results and experimental observations, as well as between AA and CG PMFs [22], they did not consider the role of the NP size and it is thus unknown how and to what extent these effective interactions are mediated by specific settings. Yet, taking into account the metal nature of Ag, we adopt the apolar (C-type) nature of NP beads from these previous Au studies [22,52], and vary the C-bead type of the NP from C1 to C5, thereby systematically changing the overall apolarity or hydrophobicity of the NP. Instead of the one- or two-component membrane of the previous studies, we consider binding to the model lung membrane.

#### 3.1.2. Potentials of Mean Force for Two NP Sizes

We find that the PMFs for both 3 and 5 nm NPs display monotonic behaviour for the most apolar C1 type—see Figure 2A,D—lacking the short-range repulsion of their atomistic counterparts. Clearly, for the most apolar or hydrophobic uniform CG NP model, the most stable situation is obtained when the NP of size 3 or 5 nm is optimally shielded from the surrounding water by inserting itself into the membrane core as much as possible. Insertion into a DPPC bilayer was experimentally observed for Ag NPs of very similar dimensions (5.7 ± 1.8 nm) that were prepared with a decanethiol coating [53]. A later study computationally reproduced this spontaneous insertion behavior using unconstrained Martini CGMD simulation for a bare 3 nm NP consisting of C1 beads [54]. Although that study considered yet another NP representation, i.e., only a shell domain with a hollow core, the finding is consistent with our computed 3 nm PMF for the uniform model with C1 beads. For completeness, we note that, in the same study, a charged 6 nm NP, generated by randomly replacing 113 C1 beads by charged Q0 beads, again only considering a shell domain and a void core, was found partially wrapped by the DPPC bilayer. We also note that the hollow core representation of an NP is just a special case of the core-shell model with vanishing LJ parameters for core beads.

Concentrating only on the insertion energy extracted from the CG PMFs, we find an overestimation by a factor of three compared to our reference atomistic PMFs for 3 and 5 nm NPs [19]. Moreover, the energetic barriers that are present in the latter at small separations correspond to the elastic penalty for the membrane to bend around and wrap the NP and we conclude from their absence in the CG PMFs that this CG representation will give rise to NPs displaying a very different translocation pathway than actually observed for bare Ag NPs at an atomistic level.

Next, focusing on a slightly less hydrophobic NP, i.e., a uniform C2 representation, the CG PMF for a 3 nm NP is strikingly similar to the atomistic PMF, except for the increase in the insertion energy; for details, see Figure 2B. Yet, for the 5 nm NP, i.e., for an NP that exceeds the membrane thickness, the CG PMF again lacks a short-range repulsion of the atomistic case and instead shows a similar monotonic behavior as for the C1 type, only with a slight decrease in the insertion energy as shown in Figure 2E.

This finding is the first clear indication that size plays a pivotal role in establishing proper NP–membrane interactions. It may both be related to the surface-to-volume ratio, which varies with NP diameter, but also to the fixed dimension of the hydrophobic core of the lipid membrane that is able to shield lipophilic NPs from the aqueous phase. We conclude that the drastic difference in the PMFs for the two considered sizes in the uniform model demonstrates the necessity of a correct CG representation for modeling NPs.

Skipping over C3, but systematically making the C-type CG bead less hydrophobic or more hydrophilic, C4 and C5, confirms this observation since the behaviour is found to reverse. For the least hydrophobic C5 type, the CG PMF for a 3 nm NP is monotonic—see Figure 3C—albeit with an insertion energy that is comparable to the atomistic case, while the PMF for 5 nm displays short-range repulsion at closer separations and shows similar behaviour to its atomistic counterparts (see Figure 2F). Since the PMFs for the C4 representation show a similar interaction profile, they are only reported in the Appendix A.

Overall, we conclude that the uniform CG NP model may be capable of properly representing individual setups, consistent with reported results in the literature [22], but that it is fundamentally incapable of capturing the essentials of NP–membrane interactions for varying NP sizes. We note that, in the past, new Martini bead types have been developed to represent NP-like carbon-based nanoparticles [20], and that we could in principle explore a similar route. Yet, rather than tuning the projection for particular cases, we may explore the separation of surface and core effects, which, as we will see further on, has the great advantage that, after calibration, no extra information will be required to perform simulation for different NP sizes.

#### 3.1.3. Exploring the Binding Mechanisms: Wrapping or Insertion

Next, we review steady state results for the uniform representation, which will offer us interesting insight into the switching of the binding mechanism in response to the NPs’ hydrophobic nature. Starting with the 3 nm C${}_{C5}$-S${}_{C5}$ case, we find the NP to completely insert itself into the hydrophobic core of the lipid bilayer, without it leading to membrane curvature; see Figure 3B. From a kinetic standpoint, after being released into the water phase, the NP moves into the hydrophobic bilayer core within 1 μs and remains there for the rest of the simulation time. For the much more hydrophobic C${}_{C2}$-S${}_{C2}$ case, a 3 nm NP remains attached to the surface and induces a slight curvature alteration to the lipid membrane, very similar to the binding characteristics observed in the atomistic system for 3 nm NPs; see Figure 3C. These results resonate with the properties of the respective PMFs discussed earlier; see Figure 2 for details. The wrapping characteristics for a stress-free elastic membrane were earlier determined from a balance of the energy required for a membrane to curve and deform, by wrapping around an NP and the energy gained in and around the contact region by replacing solvent with lipid contacts [6,7]. It should be understood that NP insertion, which is possible in practice via membrane defects and subsequent remodelling, is outside the application range of the considered elastic model. For completeness, we note that continuum models are in principle able to capture the thermodynamics of such a process [55].

In this context, it makes sense to briefly discuss findings of membrane insertion obtained by molecular simulation of a related system, α-helical peptides, which can be seen as cylindrical analogs to the small spherical NPs considered in this study. The most common rule of thumb for membrane insertion by α-helical peptides is that the free energy difference between the absorbed and inserted state should be less than 20 kcal/mol, a value that seems quite indifferent to the composition in the lipid membrane [56]. A recent coarse-grained computational study for α-helical peptides (modeled as a cylinder) with varying hydrophobic/hydrophilic surface ratios provided a more detailed understanding. It illustrated that the overall hydrophobicity and its spatial distribution are key regulatory factors for insertion. In particular, while hydrophobicity induces a transfer from an aqueous to a lipid phase, the effective hydrophobicity, and to some extent also the distribution of hydrophobic and -philic patches, will determine the height of the energetic barrier for inserting a peptide through the polar membrane exterior towards a trans-membrane orientation. Hydrophobic mismatches, which stem from incommensurability between the hydrophobic domain on the peptide and the hydrophobic core of the membrane, add to the stability of adsorbed over inserted states [57].

We may exploit these insights to discuss NP binding. When the hydrophobic mismatch is small or absent, i.e., for 3 nm NPs, increasing the hydrophobicity from C5 to C2 clearly stabilizes an adsorbed over an inserted state—see the 10 μs simulation results in Figure 3B,C—suggesting that a significant barrier for insertion is present for the uniform C${}_{C2}$-S${}_{C2}$ case, as confirmed by the PMF. The findings for the larger 5 nm NP agree very well with this interpretation and show strikingly different binding mechanisms in terms of NP wrapping versus insertion. For the C${}_{C5}$-S${}_{C5}$ case, or the least hydrophobic bead, the NP inserts and adopts a position in the center of the membrane core, albeit that the hydrophobic mismatch gives rise to membrane curvature. In particular, rather than accommodating the NP by lipids slightly bending around it as observed for the 3 nm case, lipids can be seen to form a curved monolayer around the NP. The DPPC lipids are found to move away slightly from the NP, while DOPC and POPC accumulate near the NP. This finding may be attributed to DOPC and POPC lipids having more flexible unsaturated tails, meaning that they can better adapt to curvature than their DPPC counterparts with fully saturated tails. This type of binding is somewhat reminiscent of a sketch describing DPPC membranes to which functionalized Ag NPs of roughly this size were added—see Figure 1 in Bothun [53]—albeit the sketch suggests a more straightforward membrane swelling. It should be noted, however, that the actual position of NPs in or outside the membrane remained indecisive from these measurements and that NP wrapping is consistent with the smaller liposome size fraction that was experimentally observed by dynamic light scattering (DLS) upon NP addition. Indeed, for the C${}_{C2}$-S${}_{C2}$ case, i.e., for NPs that are more hydrophobic, the 5 nm NP also inserts itself into a stable position in the center of the lipid bilayer, but this process takes place via a wrapping mechanism, in which the NP is eventually entirely encapsulated by the membrane. The lipid envelope of the NP thus initially takes the form of a bilayer, albeit that it is eventually closed up with a monolayer—see Figure 4C—which is connected to the unperturbed part of the membrane by a complicated junction region, bearing the topological features of a fusion stalk. Apparently, such a complex and curved membrane topology featuring an extended contact region between hydrophylic lipid heads and the strongly hydrophobic NP is favored over NP insertion into the membrane core via a lipid mono-layer. While this suggests the presence of a significant barrier for direct insertion, the fully wrapped NP could be a long lived metastable state. Indeed, extending the simulation by another 10 μs shows the onset of a transformation of the bilayer envelope into a lipid monolayer; see Appendix A. Concentrating on the evolution, the most hydrophobic 5 nm C${}_{C1}$-S${}_{C1}$ NP is found to move towards the center of the bilayer during the initial 500 ns, with a double bilayer wrapping around it in the next 1 μs. Interestingly, we found no pinching off for the rest of the simulation time (8 μs), so the fully wrapped state appears very long lived or even stable. The behavior for the slightly less hydrophobic C${}_{C2}$-S${}_{C2}$ case is the same; see Figure 4.

##### Stable or Metastable?

To further understand the issue of metastability, we performed an additional set of simulations using different initial conditions. We start with the fully wrapped state, i.e., the stable configuration for the most hydrophobic C${}_{C1}$-S${}_{C1}$ case, which we use as the input state for a simulation with the least hydrophobic C${}_{C5}$-S${}_{C5}$ NP. Strikingly, we observe significant undulations of the lipid membrane within the first 250 ns. Yet, as the transformation of the lipid envelope around the NP from a curved, interconnected bilayer to a monolayer will require significant remodelling, we expect the kinetics of such processes to be slow. Moreover, in the fully wrapped situation, water plays a versatile role. In particular, since the binding for more hydrophobic NPs is rather quick, a thin shell of water becomes trapped between the NP and the membrane and is subsequently co-wrapped with the NP, in full agreement with the atomistic findings. Since lipid membranes are semi-impermeable to individual water molecules, lipid monolayers only form when this continuous shell breaks up in small water clusters; see Figure 5. As a result of the interplay of curvature, the formation of water pockets and the slow bi-to-monolayer transition, unconventional membrane morphologies are formed that are all unstable on a longer time scale. The reverse case was also tested, i.e., taking the final configuration for the C${}_{C5}$-S${}_{C5}$ NP as a starting point for simulating the C${}_{C1}$-S${}_{C1}$ NP. Simulating for 10 μs, we observe no change in the monolayer around the NP, leading to the conclusion that this configuration is rather stable for the most hydrophobic NP. Combining the results, we may conclude that the state featuring a monolayer lipid coating, formed via an insertion mechanism, is energetically favoured over the bilayer coated one that originates from the wrapping mechanism; see Figure 1B. This could be corroborated by calculating the free energy difference between these two distinct states, for instance, using a string method [58], but such an exercise is outside the scope of the current study.

##### Endocytosis

Previously, Spangler et al. [7] used a combination of continuum theory and coarse implicit-solvent molecular modeling to analyze the interactions between a tensionless membrane and an NP for various NP sizes and NP–membrane surface interaction strengths. They found that increasing the interaction strength gives rise to a sequence of unbound—partially wrapped—endocytosed NPs, with critical strengths that shift to larger values with decreasing NP size. This agrees well with the consideration based on the same Helfrich model, that full encapsulation of an NP of radius *R* is purely determined by the adhesive energy density *w* and the bending rigidity $\kappa_{c}$; i.e., it is favored when $R\geq \sqrt{2k_{c}/w}$, meaning that the NP size associated with full wrapping is indeed lower bounded when considering the same membrane and NP and this size decreases with increased adhesion strength [59]. It should be noted that, within this description, pinching off into an endocytosed state takes place once NPs are entirely wrapped by a bilayer, something that was indeed found in molecular simulations using an implicit-solvent CG representation developed by Cooke and Deserno [60,61]. Our results clearly show that pinching off leading to endocytosis of a fully encapsulated NP, a process that is primarily driven by a line tension between perturbed and unperturbed parts of the membrane, is sensitive to detail and the same conclusion holds for the particular mechanism of binding. Moreover, as periodic boundary conditions may act as a constraint, care should be (and has been) taken in selecting membrane patches that are large enough to also represent the part of the membrane that is unperturbed by the binding event. Details such as the actual presence of the solvent, which may introduce a barrier for endocytosis, the lipid heterogeneity of the membrane and the finite interaction ranges can modulate the intricate balance that is responsible for the binding and unbinding mechanism and should be considered if one is interested in systematic study.

#### 3.1.4. A Core-Shell Representation for Ag NPs

Since a uniform representation is apparently incapable of reproducing key features for both 3 and 5 nm NPs, we systematically explore different combinations of all available Martini CG beads of type C in the core-shell model. While the choice for such a differentiated map may appear less intuitive than the usual mapping of equivalent atoms or atomic groups to one CG bead type, it is rather common. For instance, antifreeze beads are routinely added in CGMD simulations of aqueous systems to avoid unphysical solidification of the water phase [8]. Another example is the representation of an experimental lipid by two fractions of slightly different lipids in order to capture specific phase behaviour. After all, the overarching goal of coarse graining is to accurately represent key energetic and entropic factors that play a role in phenomena of interest at a finer level in an effective coarser description and NPs are special in this respect. Recent all-atom molecular mechanics calculations of geometrical, topological and energetic nanodescriptors for NPs composed of a broad range of materials did not only identify a natural separation of core and shell domains, but also suggested a direct link between these nanodescriptors and toxicological end points [62]. In fact, a separation into a core and shell region in a CG representation is already suggested by the different local structural properties, such as coordination and connectivity, that are observed in these domains. After all, systematic coarsening procedures, such as Iterative Boltzmann Inversion (IBI), explicitly consider structure in the form of the radial distribution functions in the inversion procedure [63]. Introducing a core and a shell particularly enables one to better match the effective surface field [64] between NPs and their environment at the atomic and coarse-grained resolution for the purpose of developing a more physically relevant CG description.

Earlier computational work already hinted that considering different CG beads for core and surface domains will affect the PMF profiles [26]. However, these studies focused on the role of distinctly different factors, such as NP shape and surface modifications, on NP binding to a more basic membrane, thereby leaving the influence of NP size (fixed to 3 nm for spherical NPs) and the nature of the hydrophobic core (fixed to C2 CG beads) untested. Moreover, being a generic study, no direct comparison was made to atomistic or experimental data. For a surface domain also composed of C2 beads, i.e., our uniform 3 nm C${}_{C2}$-S${}_{C2}$ case, the authors found a monotonic PMF corresponding to NP insertion, similar to our PMF for 5 nm Ag NP, indicating that membrane composition, which significantly differs between our and their setup, matters. On the other hand, they also found that a choice for any other of the considered (polar) surface bead types shifts the balance to membrane-adsorbed or even fully solvated NPs being more stable, thereby stressing the role of surface interactions. Based on this insight and our results for the uniform NP model, especially the recognition that the PMF for the smallest 3 nm NPs is rather accurate compared to its AAMD counterpart, we postulate that considering C2 type beads for the shell domain and CG beads of different polarity for the core domain might be sufficient to characterize the Ag NP for different sizes correctly. Such an informed limitation of the search space is useful, as the calculation of PMFs is compute-intensive even at the CG level.

##### Matching AAMD and CGMD Potentials of Mean Force

Decreasing the hydrophobicity of the core beads, by selecting the C3 type, indeed provides a distinct change in the PMF profiles particularly for the 5 nm NP; see Figure 6A,D. We identify excellent agreement with the atomistic PMF profiles for both considered NP sizes, apart from an affine translation to slightly larger separations. Yet, while the binding energy is very comparable to the atomistic reference values for 3 nm, it is overestimated by approximately 25 kcal mol${}^{-1}$ in the case of 5 nm Ag NPs. The consistent but small affine shift in the PMF profile and consequently in the position of its minimum, has been observed previously [20,22] and is attributed to the known slight mismatch between the bilayer thickness for the AA and CG cases. The finding that this shift is in the opposite direction for the two considered sizes illustrates the complexity of such a multi-dimensional fitting procedure. We attribute it to surface interactions becoming more dominant for larger NPs. Nevertheless, the important aspect of short-range repulsion due to membrane elasticity is captured at short distances; this was absent in the uniform bead NP model; see Figure 6. We conclude that the considered core-shell representation captures the main and essential features, i.e., short-range repulsion and long-range attraction between NP and the lipid membrane, and also tested even less hydrophobic core beads. Exchanging C3 core beads further by the less hydrophobic C5 type—see Figure 6B,E—does not modify the PMF profiles within the considered accuracy.

Since the relation between bead types in the core-shell representation and the resulting PMF is found to be non-linear, we also tested other combinations. A reasonable option is to test the model with the most apolar C1 beads for the shell and the most polar P1 beads for the core of the NP available within the Martini framework. Figure 6C,F illustrates that the latter choice increases the binding energy compared to the atomistic reference value for both NP sizes, despite the core being more hydrophilic than before. Moreover, while the PMF for 3 nm reproduces the position of the minimum more accurately than the other combinations, the minimum for 5 nm can be seen to shift accordingly, i.e., towards the center of the bilayer. This finding stresses again the significance of the total surface area. We conclude that, within the chosen core-shell representation, the best combination is C2 CG beads for the shell and C5 (C3 or C4) beads for the core. In particular, we find that this representation in the core-shell model captures the essential features of the PMFs that were absent in the usual uniform CG model. We will thus concentrate on this parameter set for Ag NPs in the remainder of the article.

##### Comparing Binding Mechanisms and Structural Characteristics

To further analyze the NP uptake and its effect on the complex membrane, we compared density profiles of the PO4 CG beads for pristine and NP-loaded membranes; see Figure 7 (top panel). In particular, we are interested to see whether the effects of the NP on the membrane structure found in the AAMD study are reproduced on a coarser level. With the NP size comparable to the membrane thickness, considerable deformation of the membrane is expected. The 3 nm NP is seen to distort only lipids that are in direct contact with the NP, while the 5 nm NP is seen to induce notable distortions of both leaflets. The comparison of density profiles with and without the NP clearly illustrates these differences. We also identified selective partitioning of the cholesterol in the lipid membrane. Atomistic results already showed that cholesterol is pushed away from the Ag NP and diffuses freely [35], and we observed the same behavior in our coarse-grained simulations. Figure 7 (bottom panel) shows the radial distribution functions (RDFs) for DPPC/cholesterol and the Ag NP surface. The RDFs clearly depict that cholesterol stays away from the NP in both cases.

Lipid bilayers exhibit various phases, including a liquid crystalline fluid and gel phase, depending on the temperature, degree of hydration, the cholesterol content and the lipid constituents. In complex membranes, it can also form a mixed phase and display different diffusion regimes. The smoothing of the energy landscape that is the consequence of coarsening generally enhanced diffusion rates when compared to the atomistic case. Previously, a speed up by a factor of 4 was found for the lateral diffusion of DPPC lipid at 323 K by comparing the diffusion rates extracted from the Martini CGMD with the experimentally measured diffusion rates [8]. While direct comparison with fluorescence correlation spectroscopy experiments also suggests the same speedup for PC lipids [65], this factor can actually vary with the lipid composition of the bilayer [66]. The lack of experimental data for our bilayer composition prevents the extraction of a single scaling factor for the conversion to realistic times and diffusion rates, so we will only report rates based on the considered discrete time step in Martini CGMD. Since NP binding can perturb one or both leaflets, depending on their size, the lipid pools that are considered for the reported diffusion rates are taken from both leaflets. The experimental transition temperature for DPPC membranes, i.e., a pure bilayer relating to our highest (50%) lipid content, is 314 K [67] meaning that such a membrane is in a gel phase at 310 K. Cholesterol, which is also present in our membrane in smaller (10%) quantities, is known to modulate the phase behavior of membranes to which it is added, specifically around the transition temperature for lipid phase transitions, and displays a much higher mobility and flip-flop rate. It should be understood that the Martini model, which derives its parameters from partitioning free energies in the liquid phase at a specific temperature, correctly reproduces phase transitions, but that the transition temperatures may be shifted compared to the experimental values [68].

The introduction of NPs into a lipid bilayer has been known to change the overall fluidity of the membrane. It was found that the insertion of small NPs can cause a transition from an ordered gel to a disordered gel phase [69] and silver and gold NPs were also experimentally shown to promote the fluidity of the DPPC lipid membrane to which they were added [70,71]. Our previous all-atom study corroborated that Ag NPs are able to tune the lipid mobility, particularly in the vicinity of the NP, leading to two distinct diffusion regimes depending on NP–lipid separation distance [35]. Here, we determined the lateral diffusion coefficients *D* from the slope of mean square displacement (MSD) versus time for all lipids; see Table 1. As a reference, we also added values for two uniform models of the NP. Without the NP, i.e., for a pristine lung membrane, the diffusion coefficient D${}_{DPPC}$ was determined as 3.0 × 10${}^{-7}$ cm${}^{2}$ s${}^{-1}$, or about three times faster than the experimentally known value.

From the table, we observe an interesting trend for the uniform NP models. In particular, the lipid mobility is found to always reduce upon full insertion of the NP into the bilayer or when it gets fully encapsulated, see Figure 8. Only when partially wrapped, i.e., for the 3 nm C2-C2 NP model, do we find almost the same diffusion values as for the pristine membrane, except for POPC, although the change is negligible. Apparently, with an increased fraction of lipids interacting with the inserted or fully wrapped NP, the overall lipid diffusion slows down. This is in good agreement with the earlier atomistic findings for binding Ag NPs [35]. Concentrating next on the selected C${}_{C5}$-S${}_{C2}$ CG representation for Ag NPs, the change in diffusion rates is found to vary with NP size. For the 3 nm NP, the diffusion increases quite distinctly for all the lipids, compared to the pristine case. For the 5 nm NP, all diffusion rates can be seen to decrease. Close inspection shows that the lipids in close vicinity of the 5 nm NP are slaved to the motion of the NP, which will diffuse more slowly over the membrane surface than the 3 nm NP. As such, the calculated overall diffusion rates will depend on the ratio between bound and unbound lipids, i.e., the overall membrane dimensions, meaning that these findings should not be extrapolated to different membrane areas. Our atomistic study showed that a very weakly interacting 5 nm SiO${}_{2}$ NP gives rise to a much more profound acceleration of lipid diffusion than the strongly interacting 5 nm Ag NP, corroborating that also other factors, including the flow field around the NP in the solvent phase [72,73], affect lipid diffusion. A first sign is the acceleration observed for the 3 nm Ag with a reduced wrapping area. To see whether we can again distinguish two distinct diffusion regimes, we analyze the effect of the NP on lipids that reside near and far from the NP. In this analysis, we determined the MSD versus time for a small random set of lipids that is either positioned within 0.5 nm of the NP surface or further away from the NP surface, using the trajectory after NP binding. As in the atomistic case, it clearly shows that two separate diffusion regimes are at play—see Figure 9—determined by the strong association of lipids to the NP surface.

### 3.2. Increasing Scales: CGMD for a 10 nm Ag NP

The NP sizes considered thus far, 3 or 5 nm, fall into the regime in which a continuous increase of the wrapped area with interaction strength is anticipated [37]. The same study predicted a discontinuous relation or wrapping transition for larger NPs. Typical NPs used in experiments and materials exhibit an extensive range of sizes, ranging from tiny ≤ 1 nm, to small 3–5 nm, to large 50–200 nm on a nanoscopic scale. While tiny NPs are able to permeate through the membrane, small NPs can get adsorbed or internalized based on the nature of the surface, while large NPs are known to get adsorbed onto the surface. How NPs of intermediate sizes, i.e., 5–50 nm, interact with the lipid membrane is still largely unknown. Next, we apply the CG core-shell representation of an Ag NP that we developed in this study to simulate an NP of 10 nm diameter interacting with a lung membrane. To acknowledge the longer equilibration times needed at this scale, we performed a three times longer (30 versus 10 μs) unbiased CGMD simulation. This application for larger NPs can serve a purpose for validation, since NPs in this size range are more experimentally relevant than 3–5 nm Ag NPs, which are challenged by immediate self-aggregation in water.

Figure 10 shows representative simulation snapshots along the binding pathway. At t = 0, the NP is placed 2 nm above the lipid membrane. The NP moves quickly towards the bilayer within the initial 100 ns and the membrane starts to bend around the particle, reaching a partially wrapped state at t = 250 ns; see Figure 10. The radial distribution between the PO4 and C4A CG bead of DPPC, POPC and DOPC lipids, representing lipid head and tail, respectively, with the Ag surface also depict the same behavior; see Figure 11. The slight shift in the minimum distance between the Ag surface and the C4A CG beads, compared to PO4 CG beads, demonstrates the presence of a water layer around the NP, contrary to direct contact of lipid tails with the Ag surface of the NP. In spite of the bilayer wrapping only being partial for this NP size, NP material and membrane composition, the NP eventually becomes fully engulfed by the membrane within 500 ns. In particular, a flat lipid domain appears and closes the (stable) neck region, shielding the hydrophobic NP entirely from the water. In the next stage, after 1 μs, we observe two phenomena that we identified earlier for the smaller 5 nm NP. First, the flat domain that can be seen after 500 ns develops into a curved monolayer, clearly visible around the top part of the NP. Second, the thin water shell that is present in between the lipid heads and the NP, in the region where the NP is wrapped by a bilayer, starts to break up. Water-rich domains appear first in junction voids, apparently to relieve excessive membrane curvature, since curvature is highest in junction regions. In the next stages, the monolayer slowly grows over the whole NP via orientational reshuffling of lipids, only to be counteracted by the presence of water-rich domains that are encapsulated in tiny (half) vesicles of high curvature. To further investigate the role of encapsulated water, we selected water molecules within 0.5 nm of Ag NP surface and studied their motion by calculating their mean square displacement (MSD) with time; see Figure 11. The MSD plot for the initial 0.25 μs, when the NP is moving from the solvent phase towards the hydrophobic domain of the bilayer, shows the highest slope or diffusion rate. In contrast, when the NP starts to be wrapped by the bilayer, the diffusion rates close to the NP fall by a factor of about ten. They decrease further as the NP becomes fully encapsulated, with water becoming trapped between the NP and lipid bilayer. As the hydrophobic core of the membrane in the fluid phase poses a substantial barrier for water diffusion [74], the majority of the trapped water is to be released by poration of less stable parts of the membrane. The increasing monolayer fraction and the decreasing structuring of PO4 CG beads around the NP with time further highlights the lipid reshuffling that takes place; see Figure 11. The centre of the NP also rises to a position that is level with the unperturbed part of the membrane, thereby reducing the curvature further away from the NP. Indeed, opening of a small water channel near the junction region is observed—see Appendix A—for the purpose of releasing water into the bulk, albeit most of the water remains encapsulated by vesicle-like domains further away from junctions; see the rather static structure between 10 and 25 μs in Figure 10.

Three-way junctions as shown in Appendix A have been considered before in the context of fusion pores [75,76]. It should be noted that the stability of the junction region and the role of membrane composition therein are far from completely understood. The probability of pore formation will depend on the stability of the junction and the open edge, which could lead to the detachment of the bilayer from the rest. In our case, we see that water molecules initially cluster in so-called hydrophobic ’voids’ in the junction region, an observation that is somewhat counter-intuitive, apparently in order to stabilize these domains of high curvature; see Appendix A. If we extrapolate our simulation, poration at highly curved regions may lead to a breaking up and subsequent release of an encapsulated NP. Membrane composition and lipid sorting will clearly play a role in this process. Based on the evolution, it is likely that full NP encapsulation by a monolayer is again the most stable situation. Our simulation is too short to confirm this statement, especially since the entire release of trapped water will be a long process. Although water is known to play a versatile role in molecular recognition and binding [77], our study is one of the first to investigate its function in NP–membrane interaction in such detail.

## 4. Conclusions

Despite the overall increase in the use of nanoparticles for commercial activities in the last few decades, a fundamental understanding of NP interaction with complex (cellular) membrane is still greatly lacking. In particular, NP sizes in the intermediate range (5–15 nm) evade both experimental and computational frontiers, either due to the practical limitations of experimental resolution or the diverging costs associated with spatially resolved computational studies. In this work, we developed a systematic approach for coarse graining the interactions of silver NPs with a lipid membrane. The strategy is based on a new core-shell representation of CG NPs, with core and shell consisting of Martini CG beads that may differ from each other and a matching of CG PMFs to atomistic PMFs for two NP sizes simultaneously. Although this core-shell approach was touched upon previously in the context of Martini CGMD, i.e., for representing soft coated NPs, our study is unique in that it investigates the fundamental need for a proper CG simulation setup for NPs of a distinct material like silver of different dimensions and in that it highlights the role of NP hydrophobicity (and a proper representation thereof at a coarser level) in the mechanisms of membrane binding. With this proper representation, we performed unconstrained explicit-solvent CGMD simulations of 3, 5 nm and 10 nm Ag NP, exploring the size range in which a switching of the binding mechanism is expected and approaching realistic experimental NP sizes. Our model lung membrane comprises DPPC, POPC, DOPC and cholesterol with a ratio of 5:2:2:1 and closely resembles an actual lung membrane.

In the current study, we developed an adequate CG representation or map for an Ag NP at body temperature, based on its characteristics of binding to a model lung membrane. While any map constituted between the all-atom and the CG level is only strictly valid for the thermodynamic state considered in the map, it is highly unlikely that the validity of this CG map is more restricted than general. It is therefore illustrative to note that the liquid-to-gel phase transition in a pure DPPC membrane was previously reproduced by CG Martini [68] albeit at a slightly reduced critical temperature. The adequate reproduction of characteristic dynamical properties between the AAMD and CGMD descriptions observed in this study can be seen as additional support for our map. Nevertheless, one may conclude that, while the proposed mapping procedure is general, its particulars should be tested, preferably for experimental results, if a new setup is distant from the current setup in a relevant metric, in full agreement with the usual practice in coarse-grained force-field development.

Our free energy calculations for the standard uniform model, i.e., when core and shell beads are of the same Martini type, demonstrate the inability of this frequently used CG description to accurately represent Ag NPs of different sizes. Introducing flexibility in the choice of core and shell beads (the C2 type for the shell and C5 type for the core), on the other hand, brings along transferability of the CG description with respect to NP size and enables us to reproduce intricate features, such as short-range repulsion due to NP insertion, of the all-atom description that we use as a reference. The unbiased CGMD simulations for 3 and 5 nm Ag NPs show excellent agreement with the all-atom results, both in terms of NP wrapping and membrane properties. We particularly observed the same structural and dynamic properties such as cholesterol partitioning and different diffusion regimes induced by Ag NP. These findings strengthen our confidence that the developed CG representation is more generally applicable for simulating NP–membrane interactions using Martini CGMD.

In relation to existing methods that are capable of studying membrane interaction for larger NPs based on elastic theory and very coarse molecular models with implicit solvent, where a NP can be unbound, partially or fully wrapped, the most interesting finding of this work is the recognition that partially or fully wrapped NPs may only be a metastable state. We found that lesser hydrophobic NPs insert themselves directly in the lipid membrane, accompanied by monolayer formation around it. Stronger hydrophobic NPs are partially or fully wrapped by a bilayer and may become fully encapsulated. Yet, our simulations suggest that this state is long-lived metastable and that the insertion state carrying a monolayer is the more stable one. It is likely that a change in the lipid defect density that comes with strong membrane curvature will only shift the threshold hydrophobicity, by lowering the energetic barrier that underlies the switching between these two different mechanisms. Pinching off upon engulfment is not observed in our simulations. Our findings are in line with elastic theory in the sense that the adhesion strength plays a significant role in wrapping: a uniform 5 nm NP of C1 beads is fully encapsulated by a lipid bilayer, while the same NP of less hydrophobic C2 beads is only partially wrapped. At the same time, an NP of C5 beads is internalized, a situation that is not captured by elastic theory.

The CG Martini representation that we established in this study was also used to simulate the binding mechanism of a 10 nm Ag NP using CGMD. We identified a special role for the solvent that gets trapped between the NP and membrane and was found to counteract the transition from the (wrapped) bilayer to the (inserted) monolayer structure. We believe that this study is a crucial step towards studying NP–membrane interactions for even larger NPs and/or for non-spherical NPs. The inclusion of explicit solvent, lipid heterogeneity and realistic NP sizes paves the way for more realistic simulation in understanding the toxicity effect associated with the NP inhalation. Recently, well-defined sub-15 nm bare platinum, gold and silver NPs [78] have been synthesized that can serve as a good starting point for future high-resolution validation studies of the current computational findings.

## Figures and Tables

**Figure 1 nanomaterials-12-03859-f001:**
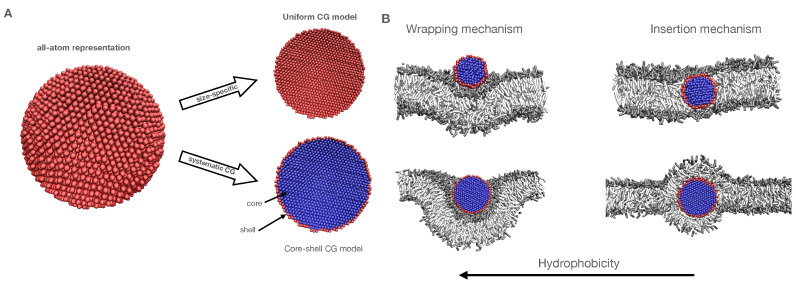
(**A**) Possible CG representations of a NP. In each case, one Ag atom is mapped to one CG bead and all surface beads make up the shell domain, while all other beads make up the core domain. In the core-shell representation, beads in shell and core domains may be of a different Martini type, while in the uniform representation, core and shell beads always are of the same type. Ag atoms are drawn red, while different bead coloring in the CGMD representation highlights the difference in Martini interaction types. (**B**) Representative CG snapshots of the two mechanisms observed in our simulations. The type is regulated by the overall NP hydrophobicity: wrapping (**B**, left) and insertion (B, right). Top and bottom rows illustrate the effect of relative size in the two mechanisms: (top) 3 nm NP, slightly smaller than the membrane thickness, (bottom) 5 nm NP, slightly bigger than the membrane thickness. As the nature of the lipids is not crucial at this stage, all hydrophobic domains of lipids are shown as white sticks, while the hydrophilic domains are shown as grey sticks. The NP representation is the same as in the core-shell CG model of panel A.

**Figure 2 nanomaterials-12-03859-f002:**
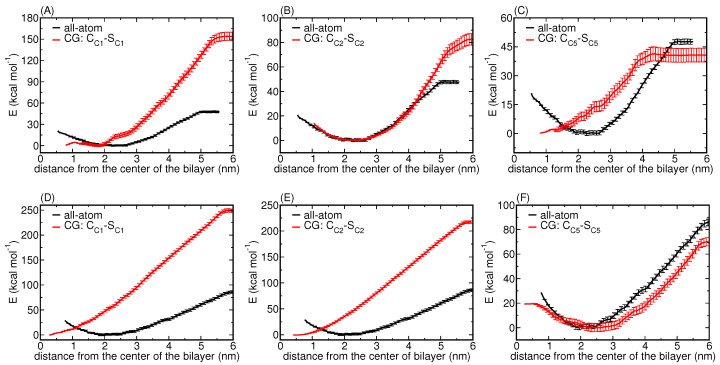
PMFs (red curves) calculated for the insertion of a CG NP of size 3 nm (top) and 5 nm (bottom) in the lipid bilayer composed of DPPC, POPC, DOPC and CHOL in a 5:2:2:1 ratio. Subplots (**A**–**F**) show CG PMFs for different CG bead types selected in the uniform model, as denoted in the legends. Although shell and core always bear the same bead type in the uniform model, the general notation of the core-shell setup is used for consistency. The shaded region corresponds to the standard deviation calculated over 100 iteration steps using the bootstrapping technique in GROMACS. The reference PMFs (black curves) were determined using constrained AAMD for the considered NP sizes in our previous atomistic study [35], and are only added here to enable a direct visual comparison.

**Figure 3 nanomaterials-12-03859-f003:**
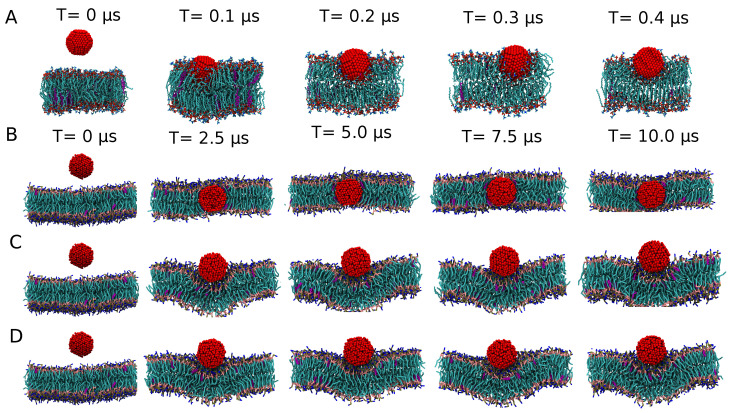
Selected snapshots showing the evolution of a 3 nm NP interacting with a lipid membrane composed of DPPC, DOPC, POPC and cholesterol in 5:2:2:1. Different resolutions and different CG bead type combinations in the core-shell model are shown: (**A**) for the reference all-atom representation of an Ag NP, (**B**) uniform C${}_{C5}$-S${}_{C5}$ CG representation, (**C**) uniform C${}_{C2}$-S${}_{C2}$ CG representation and (**D**) target C${}_{C5}$-S${}_{C2}$ CG representation of an Ag NP. As the nature of the lipids is not crucial at this stage, all lipids are shown as blue sticks and cholesterol in pink sticks, while each atom (AAMD) or bead (CGMD) in the NP is represented with a red sphere. The reference snapshots of unconstrained AAMD—see panel A—were obtained in a previous atomistic study for a smaller system [35] and are only added here to enable a direct visual comparison to the new CGMD results.

**Figure 4 nanomaterials-12-03859-f004:**
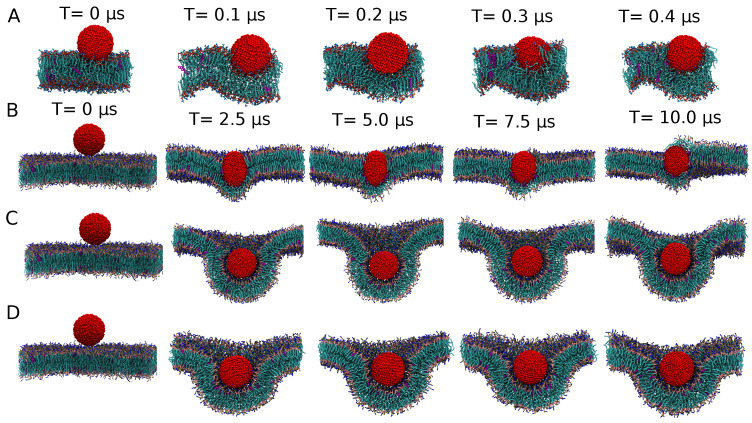
Selected snapshots showing the evolution of a 5 nm NP interacting with a lipid membrane composed of DPPC, DOPC, POPC and cholesterol in 5:2:2:1. Different resolutions and different CG bead type combinations in the core-shell model are shown: (**A**) for the reference all-atom representation of an Ag NP, (**B**) uniform C${}_{C5}$-S${}_{C5}$ CG representation, (**C**) uniform C${}_{C2}$-S${}_{C2}$ CG representation and (**D**) target C${}_{C5}$-S${}_{C2}$ CG representation of an Ag NP. As the nature of the lipids is not crucial at this stage, all lipids are shown as blue sticks and cholesterol in pink sticks, while each atom (AAMD) or bead (CGMD) in the NP is represented with a red sphere. The reference snapshots of unconstrained AAMD–see panel A—were obtained in a previous atomistic study for a smaller system [35], and are only added here to enable a direct visual comparison with the new CGMD results.

**Figure 5 nanomaterials-12-03859-f005:**
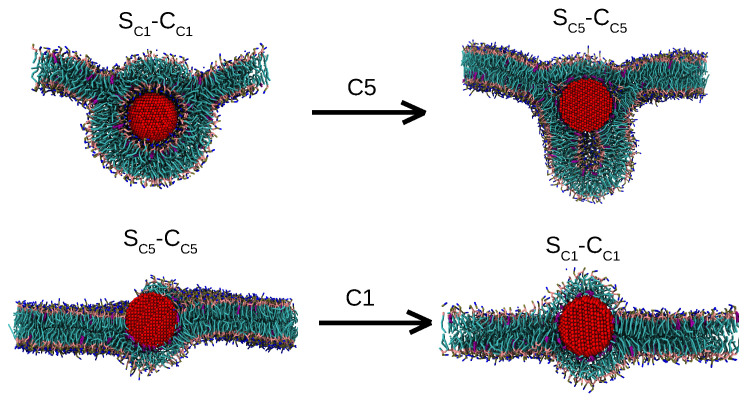
Simulation snapshots for 5 nm CG NPs showing the long-term effect of an instantaneous conversion of the NP representation in CGMD: from C${}_{C1}$-S${}_{C1}$ to C${}_{C5}$-S${}_{C5}$ (top) and from C${}_{C5}$-S${}_{C5}$ to C${}_{C1}$-S${}_{C1}$ (bottom). The structure at the end of an unbiased CGMD simulation for the original representation was used as on input for the alternative representation. As the nature of the lipids is not crucial at this stage, all the lipids are shown as blue sticks and cholesterol in pink sticks, while each CG bead in the Ag CG NP is represented by a red sphere.

**Figure 6 nanomaterials-12-03859-f006:**
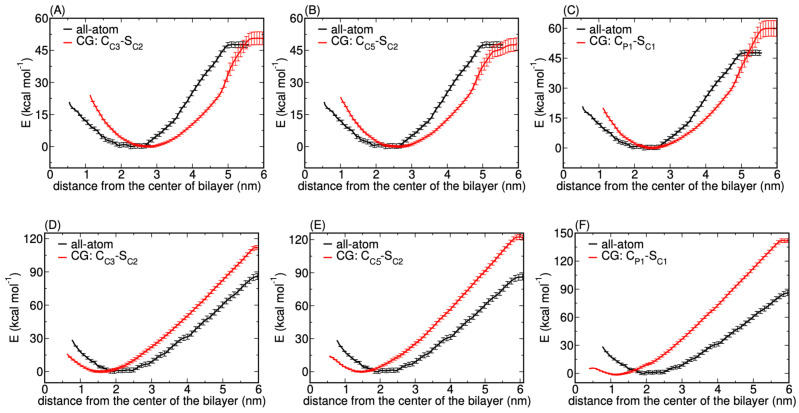
PMFs (red curves) calculated for the insertion of a CG NP of size 3 nm (top) and 5 nm (bottom) in the lipid bilayer composed of DPPC, POPC, DOPC and CHOL in a 5:2:2:1 ratio. The core-shell representation of the NP was considered. Subplots (**A**–**F**) correspond to the different CG bead combinations used to define the shell and the core of the NP. The shaded region corresponds to the standard deviation calculated over 100 iteration steps using the bootstrapping technique in GROMACS. The reference PMFs (black curves) were determined using constrained AAMD for the considered NP sizes in our previous atomistic study [35], and are only added here to enable a direct visual comparison.

**Figure 7 nanomaterials-12-03859-f007:**
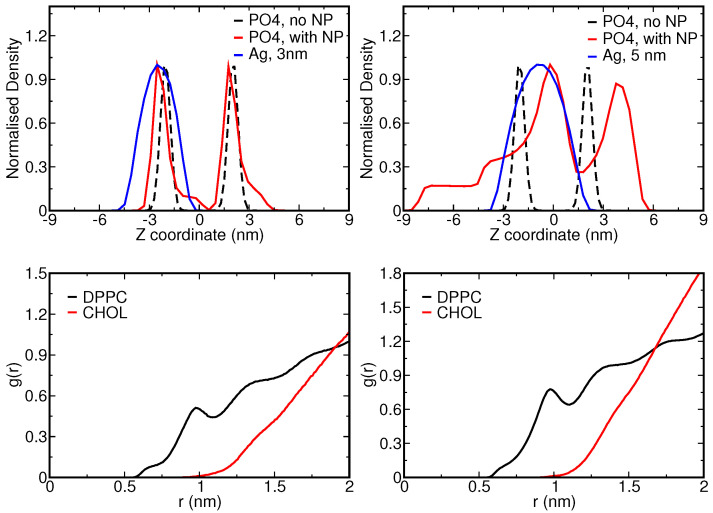
(**Top**) Normalized density profile for the PO4 CG bead of the DPPC, POPC and DOPC. Here, the black line represents the PO4 density when a CG NP is absent, the red line when a CG NP is present and the blue line represents the CG NP itself. (**Bottom**) Radial distribution functions of 3 nm (**left**) and 5 nm (**right**) Ag CG NP with cholesterol (red) and DPPC (black). In the procedure, the closest distance to the NP surface is considered.

**Figure 8 nanomaterials-12-03859-f008:**
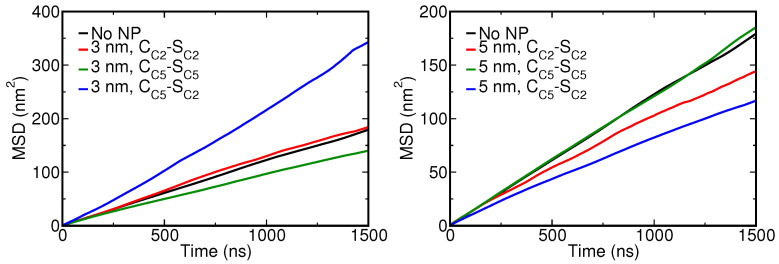
Mean square displacement vs. time for DPPC lipid in the presence of a 3 nm (**left**) and a 5 nm (**right**) Ag CG NP. The MSD was calculated for the last 2 μs of a 10 μs unconstrained CGMD simulation.

**Figure 9 nanomaterials-12-03859-f009:**
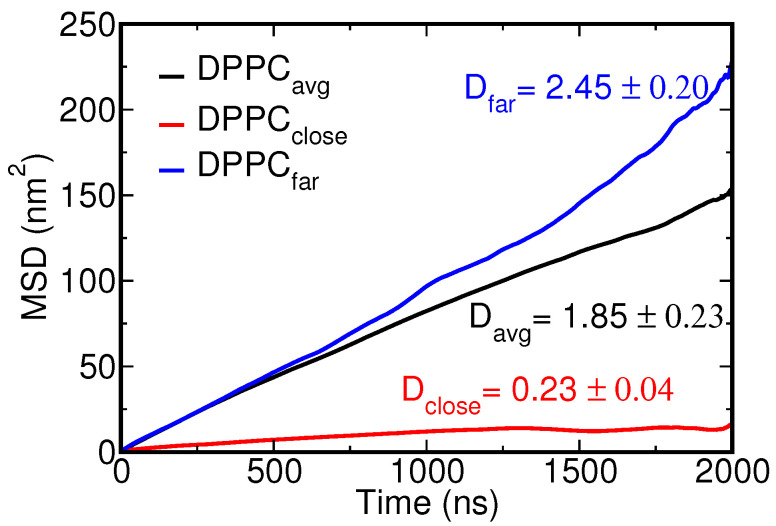
Mean square displacement (vertical axis) versus simulation time (horizontal axis) for DPPC in the presence of a 5 nm Ag CG NP. Here, DPPC${}_{avg}$ (black) refers to the average displacement of DPPC lipids—see also Table 1—DPPC${}_{far}$ (blue) refers to DPPC molecules that remain far/unbound from the NP during simulation and DPPC${}_{close}$ (red) refers to lipids that become entrapped/bound by the CG NP. The corresponding diffusion constants (in units of ×10${}^{-7}$ cm${}^{2}$ s${}^{-1}$) calculated from the MSD are added in matching colors. Diffusion rates for (un)bound lipids were determined from the last 2 μs of a 10 μs CGMD production run. Lipids in either pool, i.e., 25 bound or unbound lipids in total, were randomly selected from the two leaflets.

**Figure 10 nanomaterials-12-03859-f010:**
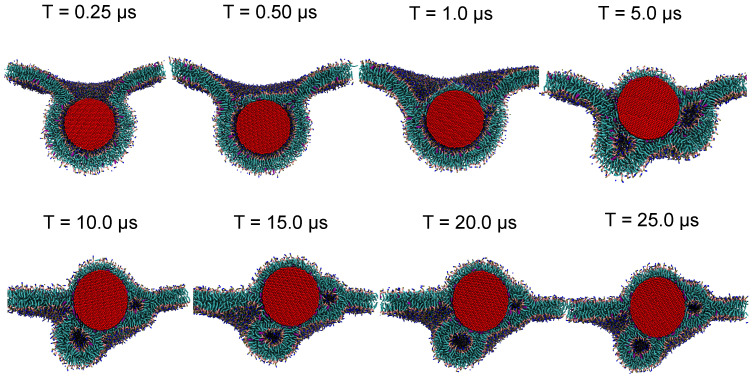
Simulation snapshots of the evolution of a 10 nm Ag CG NP interacting with a lipid membrane composed of DPPC, DOPC, POPC and cholesterol in 5:2:2:1. As the nature of the lipids is not crucial at this stage, all the lipids are shown in stick representation and cholesterol as a pink stick, while each NP CG bead is shown as a red sphere.

**Figure 11 nanomaterials-12-03859-f011:**
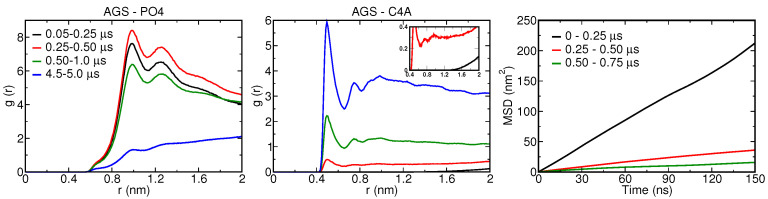
(left and middle) Radial distribution functions (g(r)) of AGS with PO4 (left) and C4A (right) CG beads averaged over various timescales. Here AGS, PO4 and C4A CG beads represent the 10 nm Ag CG NP shell and lipid head, while the C4A CG beads are the terminal CG bead of the DPPC, DOPC and POPC lipids tails. (right) Mean square displacement vs. time for CG water within 0.5 nm of the Ag CG NP surface averaged over different time windows.

**Table 1 nanomaterials-12-03859-t001:** Diffusion coefficients (in units of 10${}^{-7}$ cm${}^{2}$ s${}^{-1}$) for all lipid types in the pristine membrane and in the presence of Ag CG NP for 3 and 5 nm for different CG bead types. The MSD was computed using the last 2 μs of the 10 μs production run. The standard error for all average values is also presented.

NP Size (nm)	Bead Type (S-C)	$D_{\mathrm{DPPC}}$	$D_{\mathrm{DOPC}}$	$D_{\mathrm{POPC}}$	$D_{\mathrm{CHOL}}$
No NP	-	3.00 ± 0.00	3.26 ± 0.35	3.44 ± 0.01	3.10 ± 0.03
	C2-C2	2.97 ± 0.70	3.24 ± 0.00	4.05 ± 0.24	3.29 ± 0.01
3	C5-C5	2.28 ± 0.12	2.32 ± 0.26	2.24 ± 0.61	2.49 ± 0.37
	C2-C5	5.80 ± 0.02	5.00 ± 0.43	5.65 ± 0.30	5.83 ± 0.90
	C2-C2	2.23 ± 0.50	2.31 ± 0.90	2.62 ± 0.24	2.25 ± 0.30
5	C5-C5	3.04 ± 0.14	3.11 ± 0.11	3.00 ± 0.10	2.44 ± 0.22
	C2-C5	1.85 ± 0.23	2.03 ± 0.28	1.94 ± 0.23	1.94 ± 0.16

## Data Availability

Not applicable.

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
