# Peer review of "A Core-Shell Approach for Systematically Coarsening Nanoparticle–Membrane Interactions: Application to Silver Nanoparticles"

_nanomaterials, 2022, doi:10.3390/nano12213859_

Round 1

Reviewer 1 Report

This is an interesting and important research work concerning the relationship between silver nanoparticles and bio-membrane using modeling simulations. The authors provide sufficient background with related references. The methods are well described, and the results are clearly presented. I would recommend it be published without revision.

Author Response

We particularly appreciate the time spent reviewing our manuscript and the reports by reviewer 1.

Reviewer 2 Report

In their paper Singhal and Sevink investigate the binding of silver nanoparticles (NPs) to a model lung membrane with coarse-grained molecular dynamics simulations. This is an extension of their earlier study with all-atom simulations which they use as a reference to develop an improved CG representation of the NPs that is transferable, i.e. preserves key NP-membrane interaction features irrespective of NP size. They find that the most appropriate model for Ag NPs has 2 types of beads, one for the shell and one for the core, that differ in their hydrophobicity. Using this model they are able to reproduce the PMF profiles from corresponding all-atom simulations for 3nm and 5nm NPs and further analyze the interaction of a 10nm NP with the lung membrane in CG representation.

The paper is written well and the authors take time to provide contextual information and explain their reasoning as they take the reader through the design and execution of their simulations. The reported model and analysis represent a natural next step in their line of research on NP-membrane interactions, allowing for extension of the work to larger NP sizes and membrane patches. This study therefore would be of general interest to the scientific community in the field of nonmaterial research. 

The manuscript would benefit from clarifying the terminology describing NP-membrane interactions, providing more technical details, and addressing a few other (mostly minor) points, as outlined below. 

- Terminology. The authors describe different modes of NP interaction with the membrane including wrapping, adhesion, internalization and engulfment. Figure 1 is supposed to illustrate some of them but it is hard to visualize the key details from simulation snapshots and as a result, the description of these modes throughout the text is confusing at times. Instead, the authors should consider providing schematics of all interaction cases they describe, e.g. monolayer-wrapped vs bilayer-wrapped, adhered, internalized, engulfed, and even pinching off of the NP from the bilayer. The schematics can be numbered and the numbers used throughout the text to refer to the interaction modes including next to simulation snapshots showing specific examples from the trajectories. Part of the problem making snapshots difficult to interpret is the reduced dimensionality of the images and the inability to distinguish clearly water from lipid head groups. Having the schematics as a reference would be helpful and make it easier to follow the discussed simulation set-ups. 

- Membrane deformation energy. The authors acknowledge the contribution of the membrane deformation energy to the NP-membrane interaction mode but never discuss how the complex model of a lung membrane they use differs from a simple single-component membrane (or other model membranes) with regards to its elastic properties. Complete characterization of the mechanical constants of the membrane may be outside the scope of this study but it would be useful to know how much of the observed behavior is due to the lipid composition. For example, if cholesterol (or DPPC) concentration is increased making the membrane stiffer, would the bilayer still curve / wrap around the NP or internalize it? Are there any reasons to believe that the non-transferability of the improved NP representation would be compromised if the bilayer composition changed affecting the interaction of the NP shell with the lipids (by e.g. introducing charged lipids in the bilayer)? The authors don’t need to perform additional simulations, but a discussion on the subject would be helpful.

- Experimental validation. The earlier all-atom simulation results from the authors are used as a reference for developing the CG representation, however it remains unclear what type of experimental data on NPs exists that can be used to validate any of the simulation approaches. Even if the small size of the currently studied NPs makes their experimental measurements challenging, the presented methodological developments are intended to make the study of larger NPs more accessible. Can the authors provide some more information/references on existing experimental data that can be used now (or later for the larger NPs) to validate certain observations from the simulations? Novel findings from the simulations such as the role of water in the interaction of the 10 nm NP with the membrane are exciting, however unless some aspect of the simulation is directly related to an experimental observation, it will remain unclear whether the observed phenomena are real or artifact of the computational setup.  

- Technical details. Some technical details are missing, which would make it difficult to reproduce the approach. Please provide information on the following: the system sizes and compositions (e.g. numbers of lipids, water molecules, ions) for all simulated bilayers. Was the membrane equilibrated before placing the NP above it? Was the NP always initially positioned 2 nm above the lipid head groups, irrespective of NP size? Were any replicas run for the different simulation set-ups, in other words can the authors exclude the contribution of kinetic trapping to the observed NP-membrane interaction modes?  

Regarding the analysis: how were the density profiles and RDFs calculated from the simulations? How were the trajectories centered prior to the analysis? That is especially relevant for the density profiles of the highly curved systems. In the PMF profiles what does distance of 0 mean? Is this the case when the NP is internalized and its center of mass overlaps with the bilayer’s center of mass? Or is that distance between the NP shell and the bilayer head groups? 

For the diffusion coefficients, since the calculation involves tracing individual lipids as they diffuse laterally, how were the mean squared displacements for lipids close to and far from the NP calculated? Was the analysis confined only to lipids that remain close to (or far from) the NP during the whole trajectory (or the last portion used for analysis) and if so, what fraction of the lipids was that considering the relatively fast lateral lipid diffusion in the membrane?

- What are the advantages and disadvantages of representing the NP with a hollow core with respect to the core-shell model? The authors extensively review other NP representations but only briefly mention the particles with a void core.

- The authors note that the least hydrophobic uniform NP tends to get internalized in the bilayer while the most hydrophobic NP tends to get wrapped by the bilayer. This is counter-intuitive since the bilayer core is very hydrophobic and one would expect that a hydrophobic particle would prefer to be inside it, solvated by the lipid chains, rather than being partially wrapped by the membrane and in close proximity to lipid head groups and water. Is the barrier for insertion driving this behavior? If so, would these tendencies reverse in a membrane with more ‘defects’ that make it easier to sense/access the hydrophobic core thus lowering the barrier for insertion?

- In Figure 6 panels D, E and F, the reference PMF from the all-atom simulations shown in black is different between the plots (visibly so in D and E, and maybe also in F although the different range on the y axis makes it difficult to compare). Since that is a reference for the 5nm NP from the AA trajectories, it should be the same in all plots. It looks like it was sampled more frequently in E? Please correct this or explain the difference.

- The diffusion rates of the lipids in the CG simulations is expected to be different from experimental values due to the specifics of the force fields, as the authors mention. How do the trends in diffusion rates between lipids and close to/far from NP compare between the CG and AA simulations? Since in some cases the NP perturbs only one leaflet (e.g. Fig. 7 top left), have the authors compared the diffusion rates in lipids in the top and bottom leaflets?

- The statement that “NP rises with time” on Line 640 is ambiguous. Please clarify.

- On Lines 596-598 the authors say that typical NPs used in experiments and materials are either below 1nm, in the 3-5nm or 50-200nm ranges. It is therefore unclear what is the relevance of the intermediate regime they then focus on, i.e. the 10 nm NP. On Lines 664-665 they further point that this intermediate range evades both experimental and computational frontiers, thus validating simulation results within that range would be difficult. Are NPs in that size range commonly found/used and how can the reliability of the simulation results be assessed for them?

Author Response

We thank reviewer 2 for the extensive review and suggestions. We have answered below about the issues raised by the reviewer 2.

The manuscript would benefit from clarifying the terminology describing NP-membrane interactions, providing more technical details, and addressing a few other (mostly minor) points, as outlined below. 

  • Terminology. The authors describe different modes of NP interaction with the membrane including wrapping, adhesion, internalization and engulfment. Figure 1 is supposed to illustrate some of them but it is hard to visualize the key details from simulation snapshots and as a result, the description of these modes throughout the text is confusing at times. Instead, the authors should consider providing schematics of all interaction cases they describe, e.g. monolayer-wrapped vs bilayer-wrapped, adhered, internalized, engulfed, and even pinching off of the NP from the bilayer. The schematics can be numbered and the numbers used throughout the text to refer to the interaction modes including next to simulation snapshots showing specific examples from the trajectories. Part of the problem making snapshots difficult to interpret is the reduced dimensionality of the images and the inability to distinguish clearly water from lipid head groups. Having the schematics as a reference would be helpful and make it easier to follow the discussed simulation set-ups. 

We agree that schematics may be useful, but we are also concerned to further extend the manuscript. In particular, this information is already present in the Figures (albeit in the form of 2D projections) and in the publications (e.g. pinching off) that we refer to; moreover, the manuscript is already lengthly and the terminology is standard vocabulary in the community that considers nanoparticle-membrane interactions. To nevertheless address these concerns, we have decided to add a short explanation of the (standard) terminology when used for the first time.

  • Membrane deformation energy. The authors acknowledge the contribution of the membrane deformation energy to the NP-membrane interaction mode but never discuss how the complex model of a lung membrane they use differs from a simple single-component membrane (or other model membranes) with regards to its elastic properties. Complete characterization of the mechanical constants of the membrane may be outside the scope of this study but it would be useful to know how much of the observed behavior is due to the lipid composition. For example, if cholesterol (or DPPC) concentration is increased making the membrane stiffer, would the bilayer still curve / wrap around the NP or internalize it? Are there any reasons to believe that the non-transferability of the improved NP representation would be compromised if the bilayer composition changed affecting the interaction of the NP shell with the lipids (by e.g. introducing charged lipids in the bilayer)? The authors don’t need to perform additional simulations, but a discussion on the subject would be helpful.

We thank the reviewer for bringing this up, as it is an important point. In this study, we considered a model lung membrane containing a mixture of three different lipids (including DPPC) and cholesterol, while most previous studies have focussed on single-component lipid membranes. The lipid composition of a membrane as well as its thermodynamic state can indeed play a role in the nanoparticle (NP) translocation mechanism. This can be easily understood. From the observation that a single-component DPPC bilayer will be in the gel phase at 310 K, it is clear that it would have a significantly higher bending rigidity than when it is in the fluid phase, and thus the energy required to deform around the NP would be higher. In terms of the classical elastic (Helfrich) model, this would shift the balance between the costs of deformation/curving (defined by effective properties such as bending rigidity and lateral tension) and the NP adhesion energy (which remains constant if the membrane composition does not change). In reality, mixing in tiny amounts of cholesterol would broaden the liquid-to-gel transition region and may bring a DPPC membrane into the fluid phase at 310 K. These are thus very subtle effects, even when disregarding the role of lipid structuring. We note that the work of Lunnoo et al, referenced in the introduction, already demonstrated the impact of lipid composition on gold NP uptake. As mentioned by the reviewer, there is indeed a natural richness for further research, and it is very likely that much more work is needed to reach a definite conclusion. 

As to the bending rigidity: we focussed on reproducing the AAMD PMFs at the CG level by a newly introduced CG representation for one realistic system setup (a lung membrane at body temperature). The wrapping mechanism identified earlier and studied by continuum (Helfrich) theory only served as a starting point for our current investigation and never as a reference, meaning that the determination of effective membrane properties for our membrane (a subtle task for lipid mixtures) is not required. The main reasons for not performing a direct comparison to this continuum theory is that important mechanisms and phenomena such as lipid structuring are excluded from this continuum description, as discussed in the manuscript.  

We want to stress that there is no reason to conclude that our CG representation is only valid for the considered temperature or membrane composition. However, as noted in the introduction, any AA-CG map is only strictly valid for the considered thermodynamic state in a mathematical sense. It is, for instance, illustrative to learn that, while the liquid-to-gel phase transition in a pure DPPC membrane is reproduced by CG Martini, its critical value is reduced compared to AAMD and experiments. The adequate reproduction of characteristic dynamic properties between the AAMD and CGMD descriptions identified in this study is thus a reassuring finding. One may conclude that, while the new proposed mapping procedure is general, the particular map should be tested for appropriateness if the selected setup is far from the setup for which the map is performed (in terms of a well-chosen metric), in full agreement with the usual practice in (coarse-grained) force-field development. Future work is planned to test our current core-shell CG model for Ag by considering other NP sizes and membrane compositions, with the aim to understand the developed model's capability further.

We added a paragraph to the Conclusions in this spirit: “In the current study, we developed an adequate CG representation or map for an Ag NP at body temperature, based on its characteristics of binding to a model lung membrane. While any map constituted between the all-atom and the CG level is only strictly valid for the thermodynamic state considered in the map, it is highly unlikely that the validity of this CG map is more restricted than general. It is therefore illustrative to note that the liquid-to-gel phase transition in a pure DPPC membrane was previously reproduced by CG Martini,\cite{Biochem Biophys Res Commun 2018; 498(2):327-333} albeit at a slightly reduced critical temperature. The adequate reproduction of characteristic dynamical properties between the AAMD and CGMD descriptions observed in this study can be seen as additional support for our map. Nevertheless, one may conclude that, while the proposed mapping procedure is general, its particulars should be tested, preferably to experimental results, if a new setup is distant from the current setup in a relevant metric, in full agreement with the usual practice in (coarse-grained force-field development.”

  • Experimental validation. The earlier all-atom simulation results from the authors are used as a reference for developing the CG representation, however it remains unclear what type of experimental data on NPs exists that can be used to validate any of the simulation approaches. Even if the small size of the currently studied NPs makes their experimental measurements challenging, the presented methodological developments are intended to make the study of larger NPs more accessible. Can the authors provide some more information/references on existing experimental data that can be used now (or later for the larger NPs) to validate certain observations from the simulations? Novel findings from the simulations such as the role of water in the interaction of the 10 nm NP with the membrane are exciting, however unless some aspect of the simulation is directly related to an experimental observation, it will remain unclear whether the observed phenomena are real or artifact of the computational setup.  

As mentioned in the manuscript, the present objectives of this study are: 1) to obtain fundamental insight into the characteristics of binding with changing hydrophobicity and size, 2) to study binding for larger Ag NPs, 3) to investigate the possibility of calculating meaningful advanced descriptors for QSAR. A fourth (future) goal is validation, so we thank the reviewer for  encouraging us to validate against experimental data. 

There are actually two possible routes for this: the simulation results may be used to guide  experimental design, and concentrate on whether or not the predicted intricate phenomena are observed in experimental reality, or the simulations can be set up in line with an existing experimental design in order to evaluate if they reproduce experimental observables. Our focus is clearly on the first route, hoping that the interesting findings in this study will spur interest of experimental groups that have high-resolution imaging methods at their disposal. We note that bare sub-15 nm platinum, gold and silver NPs have been synthesised recently (see Das et al, Bare plasmonic metal nanoparticles: synthesis, characterisation and in vitro toxicity assessment on a liver carcinoma cell line, IET Nanobiotechnol. 14, 851–857, 2020), with only silver NPs showing distinct nanotoxicity at higher concentrations, and could serve as a starting point for such a validation study. The second route is more involved and requires additional work. In particular, most experimental studies for smaller NPs consider one or two-component lipid membranes, for which the binding characteristics are not necessarily directly comparable. Bothun et al, a study that was referenced in the manuscript, considered silver nanoparticles with a size distribution between 15 and 46 nm at 300K, and found that a pure DPPC membrane can accommodate these NPs by distortion, which can lead to lipid ordering and a change in their phase behavior. However, their NPs are capped/ligated, and their solution contains highly concentrated Ag NPs of different sizes (binding is size dependent). Thus, several factors should be introduced into the computational setup to sufficiently capture the current experimental reality, which is not an easy task. We have added a sentence at the end of the Conclusion section: “Recently, well-defined sub-15 nm bare platinum, gold and silver NPs\cite{Das et al, Bare plasmonic metal nanoparticles: synthesis, characterisation and in vitro toxicity assessment on a liver carcinoma cell line, IET Nanobiotechnol. 14, 851–857, 2020} have been synthesised that can serve as a good starting point for future high-resolution validation studies of the current computational findings.”

  • Technical details. Some technical details are missing, which would make it difficult to reproduce the approach. Please provide information on the following: the system sizes and compositions (e.g. numbers of lipids, water molecules, ions) for all simulated bilayers. Was the membrane equilibrated before placing the NP above it? Was the NP always initially positioned 2 nm above the lipid head groups, irrespective of NP size? Were any replicas run for the different simulation set-ups, in other words can the authors exclude the contribution of kinetic trapping to the observed NP-membrane interaction modes?  

We appreciate the concern and we sincerely apologise. We tried to be complete, as the ability to reproduce is important. As mentioned in the manuscript, the membrane area was always approximately 4.5 times larger than the NP diameter to suppress artefacts due to the periodic boundary conditions, meaning that the total system size varies with the size of the NP. A table listing volumes and compositions for each simulated system can be found in the Supplementary Information. As the reviewer assumed correctly, after equilibration of the lipid membrane patch, we manually inserted a single NP 2.0 nm away from the lipid membrane interface for all cases. To acknowledge this, the first sentence of the subsection about the setup for unconstrained MD now reads: “A single NP was manually inserted 2.0 nm away from an equilibrated lipid membrane and subsequently solvated with CG water beads (of which 5 % were antifreeze water beads).” Two replicates were ran for all constrained simulations; in addition, and as reported in the SI, we performed a convergence analysis for all PMFs. For the unconstrained case, we performed only one simulation. However, kinetic trapping was avoided by following the long-time evolution of each system, i.e. ten microseconds, as detailed in the manuscript, which was also monitored via the structural changes. The reported time scale corresponds to coarse-grained time, meaning that the effective time scale in CG is even longer, owing to the four times enhanced diffusion rate that was earlier determined for the Martini CG forcefield.

  • Regarding the analysis: how were the density profiles and RDFs calculated from the simulations? How were the trajectories centered prior to the analysis? That is especially relevant for the density profiles of the highly curved systems. In the PMF profiles what does distance of 0 mean? Is this the case when the NP is internalized and its center of mass overlaps with the bilayer’s center of mass? Or is that distance between the NP shell and the bilayer head groups? 

The density and RDFs were calculated using the built-in routines of Gromacs. We believe that relevant information concerning the followed procedures was already provided in the computational setup section. All trajectories were centred according to the NP positions, while the density profiles were centred around the lipid bilayer, to demonstrate the symmetry of the membrane. This is very helpful in easily visualising the effect of the NP on the upper and lower leaflets of the membrane. 

As mentioned in the Methods section, for the PMFs, the reaction coordinate corresponds to the center of the mass distance between the lipid bilayer and NP. Hence, the origin for the PMF plots relates to the case that the center of mass of the bilayer coincides with the NP’s center of mass. Since the dimension of the membrane is large compared to the NP size, the center of mass of the bilayer is close to that of an unperturbed membrane. 

  • For the diffusion coefficients, since the calculation involves tracing individual lipids as they diffuse laterally, how were the mean squared displacements for lipids close to and far from the NP calculated? Was the analysis confined only to lipids that remain close to (or far from) the NP during the whole trajectory (or the last portion used for analysis) and if so, what fraction of the lipids was that considering the relatively fast lateral lipid diffusion in the membrane?

As mentioned in the manuscript, two pools of lipids were made. Each pool consisted of 25 lipids, and lipids located within 0.5 nm of the NP surface were considered ‘close to the NP’ and traced for the last 2us of the 10us CGMD simulation trajectory. Their mean-square displacement was calculated using the Gromacs mid routine. A similar protocol was followed for lipids ‘far from the NP,’ i.e., more than 2 nm away from the NP surface, to determine their diffusion coefficient. We have specified the number of lipids in each pool in the caption of Fig. 9.

  • What are the advantages and disadvantages of representing the NP with a hollow core with respect to the core-shell model? The authors extensively review other NP representations but only briefly mention the particles with a void core.

In our opinion, the hollow core representation is just a subset of the core-shell model. It is a special case, where core beads do not interact with any other type of beads (i.e. for vanishing Lennard-Jones ε=0). Being just a subset in the parameter space for the core-shell model, we see no special reason to elaborate. We added one sentence after where the hollow core representation is mentioned: “We also note that the hollow core representation of a NP is just a special case of the core-shell model with vanishing LJ parameters for core beads.”

  • The authors note that the least hydrophobic uniform NP tends to get internalized in the bilayer while the most hydrophobic NP tends to get wrapped by the bilayer. This is counter-intuitive since the bilayer core is very hydrophobic and one would expect that a hydrophobic particle would prefer to be inside it, solvated by the lipid chains, rather than being partially wrapped by the membrane and in close proximity to lipid head groups and water. Is the barrier for insertion driving this behavior? If so, would these tendencies reverse in a membrane with more ‘defects’ that make it easier to sense/access the hydrophobic core thus lowering the barrier for insertion?

This is an interesting question, as our membrane is flat and tensionless, meaning that the density of defects is relatively low compared to strongly curved membrane. In our simulations, we find that the membrane partially wraps more hydrophobic NPs (C2 CG bead) and that it entirely wraps and thus encapsulates the most hydrophilic NP (C1 CG bead), fully in line with continuum theory as the adhesion energy density increases. The least hydrophobic NP (C5 CG bead) inserts itself without being wrapped. We were also puzzled by this finding at first, but understood it via the analogy to short peptides, as explained in the manuscript. Clearly, there is a barrier for direct insertion, the height of which increases with NP hydrophobicity, as the NP has to cross the hydrophilic domain formed by the lipid head groups. Owing to this barrier, the fully wrapped state (by a bilayer) is a long-lived metastable state, and the case where a monolayer forms around the NP is more stable. But this is not the complete story. For the C2 CG bead NP, the initial wrapping is only partial, but extending the simulation time reveals the formation of a monolayer to fully shield the NP from the solvent and a (very) slow transformation of the bilayer envelope into a lipid monolayer. For the C1 CG beads, i.e. the most hydrophobic NP, the driving force for the NP to shield itself from the solvent is very high, so it tends to move instantly toward the lipid bilayer. A shell of water molecules does not have time to diffuse away and gets trapped between the NP and the hydrophilic lipid head group. 

To answer the question: the height of the barrier for insertion is likely to be sensitive to the lipid density and thus to lipid defects. For tiny but very hydrophobic NPs such as C60 (0.72 nm diameter), it is known that the waiting time for C60 to enter the membrane is determined by lipid defects opening up, so there is a direct correlation. For much larger NPs, however, exposing the hydrophobic membrane interior should be collective, and it is known that the probability goes down exponentially with defect area, see the Packmem literature. So, it is likely that only the threshold where the mechanism changes from insertion to wrapping is shifted.  We added the sentence “It is likely that a change of the lipid defect density that comes with strong membrane curvature will only shift the threshold hydrophobicity, by lowering the energetic barrier that underlies the switching between these two different mechanisms.” after “…the more stable one.” in the third paragraph of the Conclusion. 

  • In Figure 6 panels D, E and F, the reference PMF from the all-atom simulations shown in black is different between the plots (visibly so in D and E, and maybe also in F although the different range on the y axis makes it difficult to compare). Since that is a reference for the 5nm NP from the AA trajectories, it should be the same in all plots. It looks like it was sampled more frequently in E? Please correct this or explain the difference.

We have incorporated plots with same format of error in the modified manuscript.

  • The diffusion rates of the lipids in the CG simulations is expected to be different from experimental values due to the specifics of the force fields, as the authors mention. How do the trends in diffusion rates between lipids and close to/far from NP compare between the CG and AA simulations? Since in some cases the NP perturbs only one leaflet (e.g. Fig. 7 top left), have the authors compared the diffusion rates in lipids in the top and bottom leaflets?

The diffusion rates at the CG level are about three times faster than the experimental and our all-atom results. The all-atom diffusion rate increased almost two-fold for the lipids away from the nanoparticle. At the same time, it decreased by at least two orders of magnitude for lipids in the vicinity of the nanoparticle. The approximately same trend is observed in coarse-grained simulation: the diffusion rate show a 1.5 times increment for lipids that are away from the nanoparticle, while only an order of magnitude slowing down was observed for lipids within 0.5 nm of the nanoparticle. However, these diffusion rates cannot be compared to experiments, owing to a lack of experimental data for our bilayer, let alone for individual leaflets. It even forbids the extraction of a single scaling factor for the conversion to realistic times and diffusion rates. 

Overall, binding NPs can perturb one or both leaflets depending on their size, so for generality the diffusion rate is averaged over both leaflets. The random subset of lipids selected for the further distinction of diffusion regime includes lipids residing in both leaflets. So, the reported diffusion rate shows the average effect on lipids by the NP. We clarified this in the part of the text where we discuss diffusion rates, i.e. after “…so we will only report rates based on the considered discrete time step in Martini CGMD.” by “Since NP binding can perturb one or both leaflets, depending on their size, the lipid pools that are considered for the reported diffusion rates are taken from both leaflets.” 

  • The statement that “NP rises with time” on Line 640 is ambiguous. Please clarify.

We meant that the midpoint of NP first lies substantially below the average membrane position and then rises up. We changes the description to “The centre of the NP rises to a position that is at level with the unperturbed part of the membrane.”

  • On Lines 596-598 the authors say that typical NPs used in experiments and materials are either below 1nm, in the 3-5nm or 50-200nm ranges. It is therefore unclear what is the relevance of the intermediate regime they then focus on, i.e. the 10 nm NP. On Lines 664-665 they further point that this intermediate range evades both experimental and computational frontiers, thus validating simulation results within that range would be difficult. Are NPs in that size range commonly found/used and how can the reliability of the simulation results be assessed for them?

This issue is addressed in the response to an earlier question, so no new text will be added. NPs in the lower range, 1-3 nm, are in the focus of detailed computational studies due to cost limitations, while experiments consider only NPs in the range of 50-200 nm due to limitations in resolution. In nature and in commercial products, however, NPs have various shapes and sizes, so a more general understanding is needed. The intermediate regime is of high interest to the community, since membrane adhesion characteristics are seen to change dramatically between 1-50 nm. Furthermore, generating additional data points by CG methodology in this range may enable us to extract fundamental rules and extrapolate results for data-driven methodologies and to more fundamentally understand the effect of NP size on binding and translocation over a complex lipid membrane. We note that even at the CG resolution, simulating 10 nm NP interactions with a complex lipid membrane is costly and time-consuming. This work provides an appropriate representation of Ag NPs at the CG level and paves a way towards simulating even much bigger and experimentally more realistic NPs. 

Reviewer 3 Report

Dear Editor, 

this is not strictly my research field, this is the reason why I needed some more time to properly read the manuscript. I found the manuscript clear enough also for not expert reading, and readable. 

The possibility to model the interaction of NP with bio membranes represents a valuable research field owing the bio-medical relevance of Ag-NPs. The method presented by the Authors seems correct and promising. The results are consistent and valuable. 

Author Response

We particularly appreciate the time spent reviewing our manuscript and the reports by reviewer 3.